# Shear Failure Mode and Concrete Edge Breakout Resistance of Cast-In-Place Anchors in Steel Fiber-Reinforced Normal Strength Concrete

**Jong-Han Lee** [1] , **Eunsoo Choi** [2] **and Baik-Soon Cho** [3,*]

1    Department of Civil Engineering, Inha University, Incheon 22212, Korea; jh.lee@inha.ac.kr
2    Department of Civil Engineering, Hongik University, Seoul 04066, Korea; eunsoochoi@hongik.ac.kr
3    Department of Civil and Urban Engineering, CTRC, Inje University, Gimhae 50834, Korea
*    Correspondence: civcho@inje.ac.kr

**Abstract:** Concrete edge failure of a single anchor in concrete is strongly dependent on the tensile performance of the concrete, which can be greatly improved by the addition of steel fibers. This study investigated the effect of steel fibers on the shear failure mode and edge breakout resistance of anchors installed in steel fiber-reinforced concrete (SFRC) with fiber volume percentages of 0.33, 0.67, and 1.00%. The anchor used in the study was 30 mm in diameter, with an edge distance of 75 mm and embedment depth of 240 mm. In addition to the anchor specimens, beam specimens were prepared to assess the relationship between the tensile performance of SFRC beams and the shear resistance of SFRC anchors. The ultimate flexural strength of the beam and the breakout shear resistance of the anchor increased almost linearly with increasing volume fractions of fiber. Therefore, based on the ACI 318 design equation, a term was proposed using the ultimate flexural strength of concrete instead of the compressive strength to determine the concrete breakout shear resistance of an anchor in the SFRC. The calculated shear resistance of anchors in both the plain concrete and SFRC were in good agreement with the measurements. In addition to the load capacity of the SFRC anchors, the energy absorption capacity showed a linear increase with that of the SFRC beam.

**Keywords:** anchor; shear behavior; concrete edge breakout resistance; ultimate flexural strength; energy absorption capacity; steel fiber

## 1. Introduction

Concrete anchors are commonly used to support structural members and equipment in civil and industrial structures, including power plants. In addition, anchor systems are used to connect new structural members for strengthening and retrofitting existing structures [1]. In particular, large-scale structural members, facilities, and equipment are connected using pre-installed cast-in-place (CIP) anchors [2]. The failure of anchors installed in concrete is mainly dependent on the strength of the steel anchor and concrete. Concrete failure causes sudden destruction of an anchor system, which can directly affect the proper performance of structures and human safety. Thus, the evaluation of concrete fracture strength in the anchor system is essential for the stability and durability of a structure.

Studies on the concrete edge breakout strength of anchor bolts in concrete have mainly focused on anchors installed in non-reinforced plain concrete. For single and multiple anchors in plain concrete, design equations were developed based on the experimental results and discussions of previous researchers [3–5] for the concrete breakout strength of anchors using the concrete capacity design (CCD) method. This method assumes that the angle of the breakout cone shape is 35 degrees. The CCD method has been adopted in the current design standards [6–9] to determine the concrete breakout capacity of

anchors installed in concrete. Olalusi and Spyridis [10] derived a statistical model from the database of anchor tests for the concrete breakout capacity of single anchors in shear, and they showed that the CCD model was highly scattered and biased. The CCD method is a semi-empirical design method largely dependent on the test data. For limited data of concrete cone failure, Bokor and Sharma [11] evaluated load and displacement behavior of anchor groups in tension as the basis for development of concrete cone failure of anchorage with various aspects. In addition, Pürgstaller et al. [12] applied concrete anchors to nonstructural components and investigated the hysteresis shear behavior of the anchors. Dengg et al. [13] assessed the applicability of anchor systems using tunnel excavation material.

Concrete has very weak tensile capacity, so steel bars are used to support the tensile force generated in the concrete. The ACI 318M-08 [8] has proposed modification factors to increase the concrete breakout strength of anchors installed in concrete reinforced with steel bars. However, experimental studies on the effect of the reinforcement on the resistance of anchors are still limited. Moreover, when using steel bars in concrete, careful attention should be paid to the placement and corrosion of the bars [14]. Thus, many studies have also been carried out to improve the tensile capacity and ductility of cement-based materials using discrete short-length fibers, such as steel, carbon, textile, and natural fibers. In particular, steel fibers with great strength and ductility have been actively studied. For the application of steel fibers in cement-based materials, pullout tests have been performed to evaluate the bond and slip mechanisms between the steel fibers and cement-based matrix, as well as the effects of the fibers on bridging and arresting crack propagation and opening [15–17]. The addition of steel fibers enhances flexural capacity in concrete beams, and the increase in the flexural capacity is proportional to the fiber content [18,19]. In addition to the flexural behavior of steel fiber-reinforced concrete (SFRC), Narayanan and Darwish [20], Sharma [21], and Amin and Foster [22] applied steel fibers to increase the shear strength of concrete beams. Khuntia et al. [23] assessed the increase in the shear strength of concrete with steel fibers in the normal and high-strength concrete matrix. Furthermore, the influence of the concrete strength on the strength and ductility of the SFRC concrete has been studied. Mansur et al. [24] indicated the improvement of strength with steel fibers in high-strength concrete, and Holschemacher et al. [25] investigated the effect of fiber type and content in high-strength concrete. Recently, Lee [26] evaluated the influence of the matrix strength of concrete on the post-cracking residual strength of SFRC. With the contribution of previous studies to the application of steel fibers in concrete materials and structures, SFRC is currently used reliably in practical applications, such as industrial slabs, pavements, tunnel shotcrete, and precast tunnel segments [27–30].

However, the application of SFRC to anchor systems in concrete is very limited. Most previous studies have focused on anchors in non-reinforced plain concrete, and some studies have been concerned with anchor systems in steel bar-reinforced concrete. Thus, the current design equations only provide the concrete resistance of anchors installed in plain concrete or steel bar-reinforced concrete. Recently, for an anchor in SFRC, Nilforoush et al. [31] evaluated the breakout concrete capacity of an anchor bolt in SFRC members under tensile load and showed a great increase in the tensile resistance and toughness of the anchor. Tóth and Boker [32] investigated the behavior of anchorages in SFRC and showed increases in concrete breakout capacity and displacement due to the presence of fibers. Mahrenholtz et al. [33] performed experiments on anchor channel bolt systems in plain and fiber-reinforced concrete to develop the basis of the design rules for fasteners installed in FRC. Lee et al. [34] performed shear tests of SFRC anchors to assess the relationship between the mechanical properties of SFRC and the shear resistance of anchors embedded in SFRC. Then, they proposed a modified design equation that can calculate the concrete breakout strength of anchors in plain concrete and SFRC using the equivalent flexural strength ratio.

SFRC exhibits similar compressive strength to plain concrete but a significant difference in tensile capacity. Thus, this study aims to expand knowledge on anchors in SFRC and replace the compressive strength of concrete with the tensile capacity, which can be more simply and generally employed for concrete anchors than the equivalent flexural strength ratio. For this, this study performed shear tests for anchors 30 mm in diameter with an edge distance of 75 mm and embedment depth of

240 mm. The effect of steel fibers on the shear failure mode and breakout resistance of the anchors was investigated. The fiber volume percentages were changed from zero to 1.00%, and the failure mode was investigated to determine the cracking resistance, ultimate shear resistance, and energy absorption capacity of the anchors.

The increases in the concrete breakout strength and ductility of the anchor system are strongly associated with the tensile performance of the concrete. Thus, beam specimens were also prepared to assess the relationship between the tensile performance of SFRC beams and the shear resistance of SFRC anchors. The ultimate flexural strength of the beam and breakout shear resistance of the anchor increased almost linearly with increasing fiber volume fractions. Thus, based on the ACI 318 design equation, a term is proposed using the ultimate flexural strength of concrete instead of the compressive strength to determine the concrete breakout shear resistance of the anchors. In addition, this study proposes a relationship between the energy absorption capacity of the SFRC anchors and that of the SFRC beams. The results of the proposed equation showed good agreement with the experimental values.

## 2. Experiments

### 2.1. SFRC Material

The concrete used in the study was provided by a ready-mixed concrete company, and its compressive strength was designed as 27 MPa. The water-to-cement ratio was 0.54, and the slump value was 150 mm. The mix proportion of the concrete is given in Table 1. All of the materials except the steel fibers were mixed in a ready-mixed concrete truck for around 30 min until reaching the site where specimens were manufactured. The steel fibers were then added into the truck at the site and mixed with the concrete mixture. The steel fibers used in the study had hooked ends that are commonly used in practice. The length and diameter of the fibers were 60 and 0.75 mm, respectively, and thus the aspect ratio was 80. The ultimate tensile strength of the fiber was 1100 MPa.

**Table 1.** Mix proportion of concrete (kg/m$^3$).

| Cement | Fine Aggregate | Coarse Aggregate | Fly Ash | Superplasticizer |
|--------|----------------|------------------|---------|------------------|
| 279 | 931 | 929 | 31 | 1.17 |

The compressive strength of concrete was measured using cylinder specimens with a diameter of 10 mm and length of 20 mm. The average compressive strength of the non-reinforced plain concrete measured from four cylinder specimens was approximately 28 MPa. The average compressive strengths of the SFRC with fiber volume percentages of 0.33, 0.67, and 1.00%, measured from four cylinders for each fiber volume fraction, were in the range of 26 to 28 MPa. Thus, the measured compressive strength showed very good agreement with the design strength of 27 MPa. Little difference was found between the non-reinforced and SFRC, which means that there was little influence of the steel fibers on the compressive strength of the concrete.

### 2.2. Preparation of the Anchor and Beam Specimens

The concrete breakout failure of an anchor installed in concrete is strongly associated with the tensile performance of concrete. Steel fibers mixed in concrete have a great influence on the generation and growth of concrete cracks. Thus, to investigate the effect of steel fibers on the concrete edge breakout failure of an anchor, the anchor specimens of SFRC with fiber volume percentages of 0.33, 0.67, and 1.00%, which correspond to approximately 26, 54, and 80 kg/m$^3$, respectively, were designed, as shown in Table 2. The steel anchor used in the study was an M30-S45C with a diameter of 30 mm and yield strength of 450 MPa. One steel anchor was installed in the center of each edge of a concrete block. Thus, four anchor specimens per test variable were prepared in one concrete block. One concrete

block without steel fibers was also prepared, and a total of 16 anchor specimens were fabricated (four anchors in one concrete block per test variable).

**Table 2.** Specimen names and numbers according to fiber volume fractions.

| Name of Specimens | Number of Concrete Blocks | Number of Anchor Specimens | Fiber Volume Percentages, $v_f$ (%) |
|---|---|---|---|
| V000 | | | 0.00 |
| V033 | 1 | 4 | 0.33 |
| V067 | | | 0.67 |
| V100 | | | 1.00 |

To induce the concrete edge breakout failure of an anchor, the edge distance of the anchor was defined as 75 mm (=2.5 times the diameter of the anchor). The embedded depth of the anchor in concrete was 240 mm, which was 8 times the diameter of the anchor. Thus, the concrete block was designed to have a square top surface with a length of 500 mm so that the inclined cracks generated on the surface of the block due to the shear load could not affect the other edge faces. The height of the block was defined as 680 mm in consideration of the embedded depth of 240 mm of the anchor. Figure 1 shows the dimensions of the concrete block and anchor specimens. The concrete blocks with steel anchors were cured in air for around 90 days.

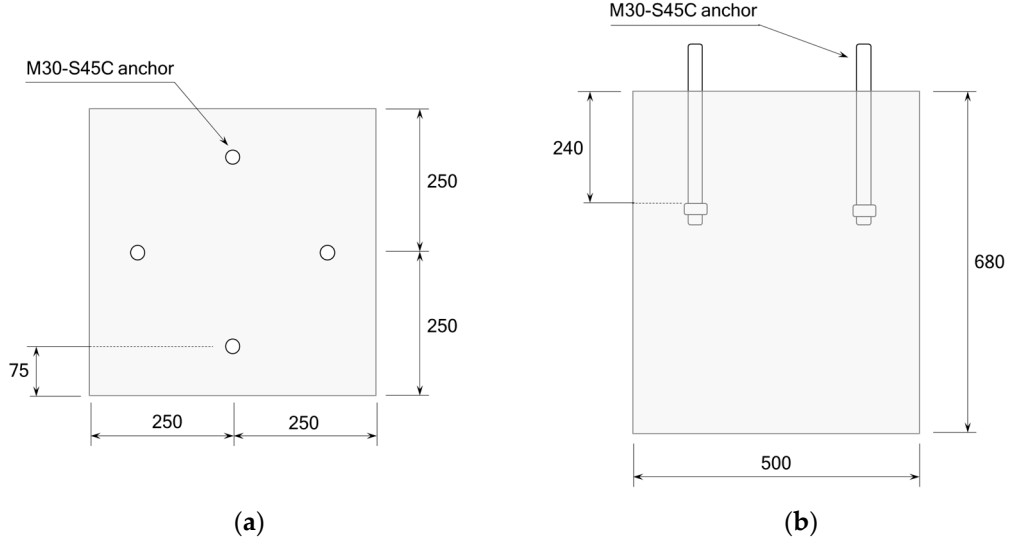

(**a**)          (**b**)

**Figure 1.** Dimensions of the anchor specimen in concrete: (**a**) top surface; (**b**) side surface (units: mm).

Beam specimens were also manufactured to assess the tensile capacity of the SFRC according to the ASTM C 1609 standard [35]. The dimensions of the beam specimens were $150 \times 150 \times 500$ mm$^3$. The beam specimens were cast at the same time as the anchor specimens using the same concrete mixture and fiber volume fractions. Control beam specimens without steel fibers were also prepared. The same number of beam specimens as anchor specimens was prepared to ensure the reliability of the experiments (four specimens per fiber volume fraction). Thus, 16 beam specimens were manufactured in total. After de-molding, the beam specimens were cured in air under the same conditions as the anchor specimens.

### 2.3. Installation and Measurement of the Specimens

The concrete anchor blocks were supported using steel angles at the four corners of the front and back sides at the bottom of the block, as shown in Figure 2a. To prevent the reaction force of the steel angles from affecting the shear behavior of the anchor, the contact width and height of the steel angle with the anchor block were limited to 100 and 400 mm, respectively. Four steel angles were

fixed to holes in the laboratory floor using high-strength bolts and nuts. Rubber plates were installed between the concrete surface and the steel angle to minimize the influence of frictional force and stress concentration on the anchor block.

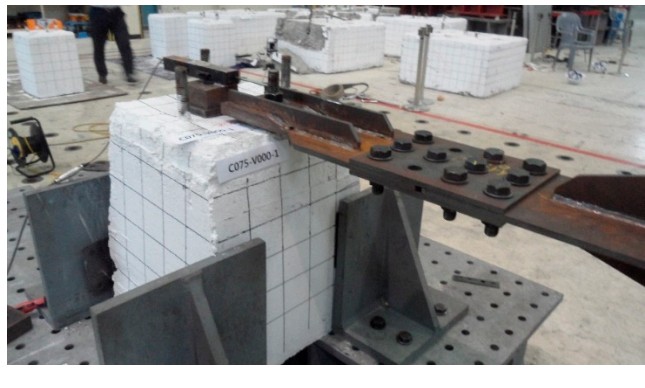

(**a**)

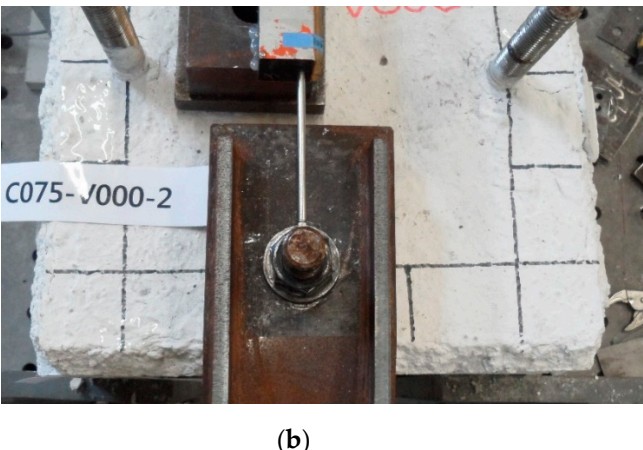

(**b**)

**Figure 2.** Test setup and instrumentation of the anchor specimen: (**a**) shear test setup; (**b**) linear variable differential transformer (LVDT) installed on the anchor.

To apply a monotonic shear load to the anchor installed in the concrete block, a loading plate 150 mm wide and 20 mm thick with a hole at the end of the plate was connected to the anchor bolt, as shown in Figure 2b. To avoid contact between the loading plate and the top surface of the anchor, a washer with a thickness of 2 mm was inserted into the anchor bolt before the loading plate was installed. Then, a nut was used to loosely connect the loading plate and anchor bolt to prevent the loading plate from moving during the experiment and to maintain a pin connection. The other end of the loading plate was connected to an actuator with a capacity of 200 kN. The actuator was installed on a fixed frame to maintain the horizontal condition during the test. The applied load was monitored with a load cell installed at the actuator. The displacement of the anchor was recorded using a linear variable differential transformer (LVDT) installed on the back of the anchor, as shown in Figure 2b. The test was continued until concrete breakout failure completely occurred.

The beam specimen was installed on a steel roller under a simply supported condition. The distance between the supports was 450 mm, and the cast top surface was used as the side surface of the beam. To distribute the supporting force evenly through the width of the beam, a rubber plate which was 30 mm in width and 3 mm in thickness was placed at the contact portion between the steel roller and beam. Two vertical loads were then applied at a rate of 0.2 mm/min on the top surface of the beam to generate a pure moment between the two loads. A rubber plate was also installed between the point of the loading and the top surface of the beam to prevent local cracks in the concrete at the loading points and to allow the uniform distribution of the load.

The bending test was continued for around 15 min until the deflection of the span reached 1/150 of the span length at mid-span. The vertical deflection was measured using LVDTs installed at the front and back sides of the beam at mid-span. Gopalaratnam et al. [36] reported that the deflection measured at the middle of the beam could be approximately twice the real deflection due to additional deflections caused by the elastic and inelastic behavior of the loading device and the slip of the specimen. Thus, a lateral frame was fabricated and attached to the beam according to the recommendation of ASTM C 1609 [35] to exclude the additional deflection and to measure deflections at mid-span.

## 3. Bending Test Results and Discussion

### 3.1. Tensile Strength before and after Cracking

According to the ASTM C 1609 [35], the stress $f$ can be calculated by

$$f = \frac{PL}{bd^2} \tag{1}$$

where $P$ is the applied load, $L$ is the span length, and $b$ and $d$ are the width and depth of the beam, respectively. Figure 3 shows the relationship between the cracking stress $f_{cr}$ and ultimate stress $f_u$ and the steel fiber content. The $f_{cr}$ corresponds to the first peak stress defined in the ASTM C 1609 [35] and the stress at the limit of proportionality defined in the BS EN 14,651 [37]. $f_{cr}$ is equal to $f_u$ in the non-reinforced plain concrete beam, which exhibited no strength recovery greater than $f_{cr}$ after cracking. The linear lines in Figure 3 were obtained from the linear regression analysis of the stress with respect to the steel fiber volume percentages from zero to 1.00%. $f_{cr}$ is almost constant, while $f_u$ increases in proportion to the steel fiber content. $f_u$ was approximately 7.40 MPa for the reinforced concrete beam with a fiber volume percentage of 1.00%, which was 2.21 times that of the non-reinforced plain beam.

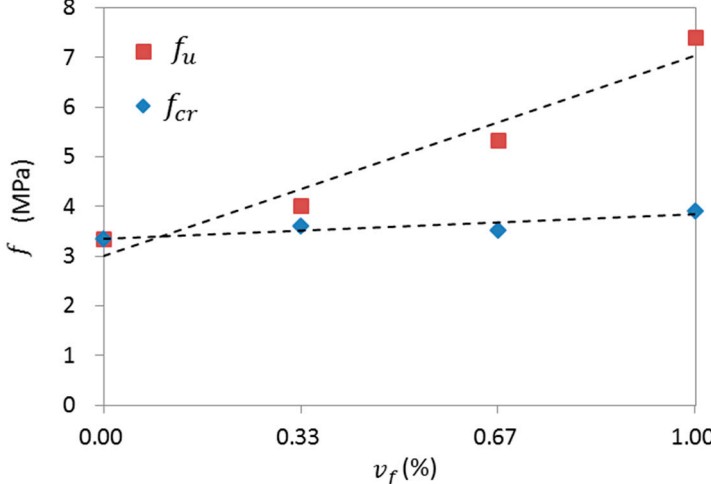

**Figure 3.** Cracking and ultimate strengths with increasing fiber volume percentages from zero to 1.00%.

The residual tensile strength is defined as the stress at the deflection of L/600 (=0.75 mm) and L/150 (=3.0 mm) according to the ASTM C 1609 [35]. JSCE SF-4 [38] defines the residual strength as the mean stress from the beginning (=zero deflection) to the deflection of L/150. Lee et al. [14] excluded the uncracked region to calculate the residual strength up to a certain deflection. In this study, the method proposed by Lee et al. [14] is used to define the residual strengths of $f_{600}^{eq}$ and $f_{150}^{eq}$, which are the mean stresses from the crack initiation to the deflection of L/600 and L/150, respectively.

Figure 4 shows $f_{600}^{eq}$ and $f_{150}^{eq}$ with respect to the steel fiber volume fractions. The linear lines shown in Figure 4 were obtained from the linear regression analysis, except for the residual strength of the non-reinforced concrete beam, which was zero. The results of the linear regression analysis show

that $f_{600}^{eq}$ and $f_{150}^{eq}$ increase in proportion to the content of steel fiber. $f_{150}^{eq}$ is slightly greater than $f_{600}^{eq}$, which means that the steel fibers play an effective role in increasing the residual strength. The $f_{150}^{eq}$ of the reinforced beams with fiber volume percentages of 0.33, 0.67, and 1.00% was 3.96, 4.83, and 6.66 MPa, which are approximately 110, 136, and 170% of $f_{cr}$, respectively.

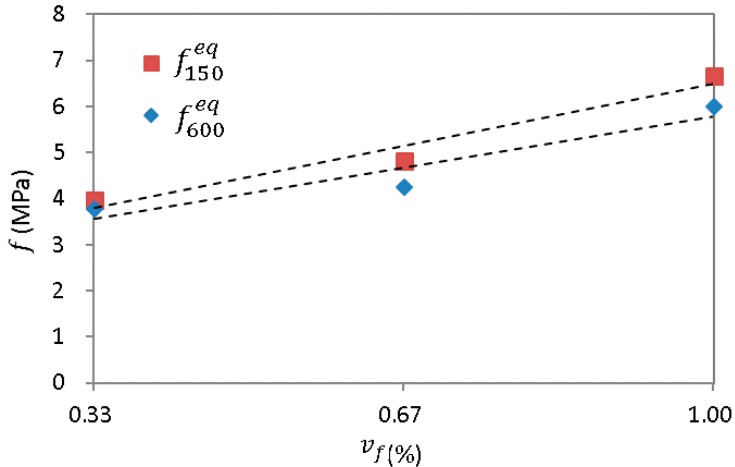

**Figure 4.** Residual strengths $f_{600}^{eq}$ and $f_{150}^{eq}$ with increasing fiber volume percentages from 0.33 to 1.00%.

### 3.2. Energy Absorption Capacity

The energy absorption was calculated from the area of the stress and deflection curves obtained from the bending tests. Figure 5 shows the relationship between the deflection and energy absorption of the beams. In the plain concrete beam, the residual strength was very small after cracking, and the increase in the energy absorption capacity was minimal with increases in deflection. However, the energy absorption capacity of the SFRC beams increased almost linearly with increasing fiber content.

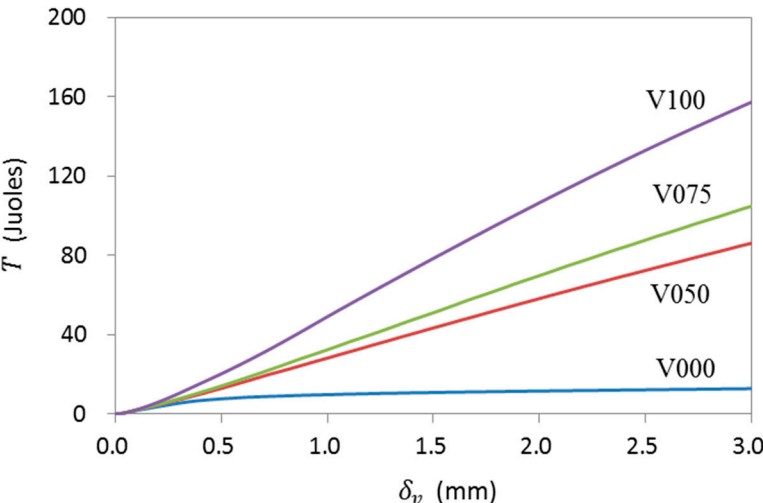

**Figure 5.** Energy absorption and deflection curves of the plain concrete and steel fiber-reinforced concrete (SFRC) beams obtained from the bending tests.

Figure 6 shows the energy absorption capacities of $T_{600}$ and $T_{150}$ with increasing fiber volume fractions. $T_{600}$ and $T_{150}$ are the energy absorption capacities at deflections of $L/600$ (=0.75 mm) and $L/150$ (=0.0 mm), respectively. The linear lines obtained from the regression analysis with respect to the fiber volume percentages from zero to 1.00% show that $T_{600}$ and $T_{150}$ increase proportionally to the increase in the steel fiber content. The slope of the regression line of $T_{150}$ is greater than that of $T_{600}$, which means that the steel fibers effectively maintained greater residual strength than the cracking

strength until the deflection of $L/600$. For the concrete beam with a fiber volume percentage of 1.00%, $T_{600}$ and $T_{150}$ were 31.02 and 143.72 J, which correspond to 3.47 and 11.25 times the values of the non-reinforced plain beams, respectively.

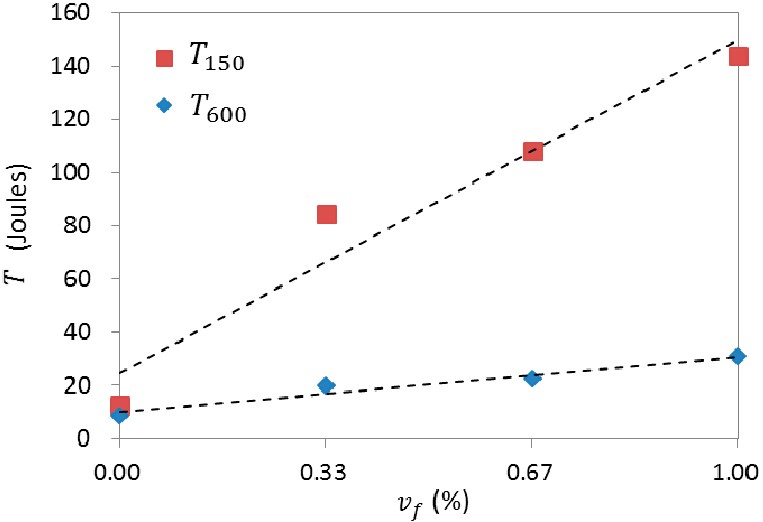

**Figure 6.** Energy absorption capacities $T_{600}$ and $T_{150}$ with increasing fiber volume percentages from zero to 1.00%.

## 4. Shear Test Results and Discussion

### 4.1. Test Results and Failure Mode

Figure 7 shows the load and displacement curves measured in the shear tests of the anchors installed in the plain and SFRC. The load and displacement curve can be divided into pre-cracking and post-cracking regions based on a point when a concrete crack occurs on the top surface of the concrete block. Before the crack occurs, the tensile stress generated in the concrete block is below the cracking strength of the concrete. Thus, all of the materials including concrete linearly and elastically resist the external shear loads, and the load and displacement curve of the anchor in the SFRC is similar to that in the non-reinforced plain concrete, as shown in Figure 7.

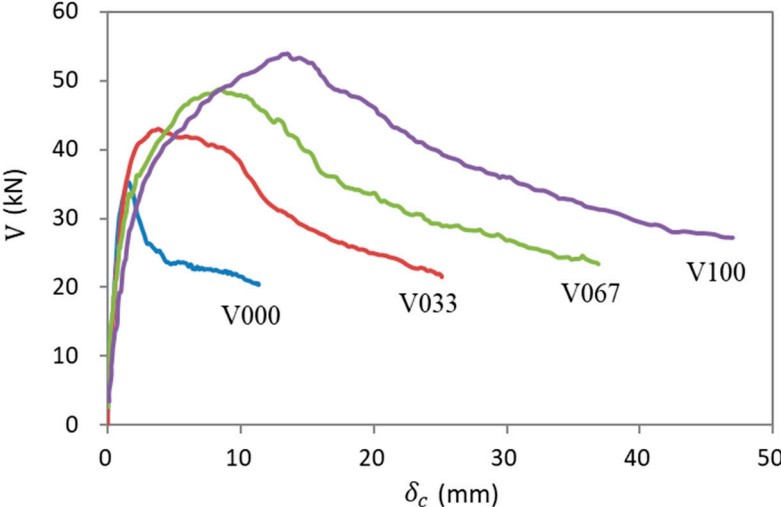

**Figure 7.** Typical load and displacement curves measured from the shear tests of anchors in the plain concrete and SFRC.

When a crack occurred on the top surface of the concrete block, the slope of the load and displacement curve decreased due to the strength reduction of the concrete. Cracks initiated almost simultaneously on the left and right sides of the steel anchor on the top surface of the concrete block. The left and right cracks occurred at an angle on the top surface and proceeded to the edge of the block. The inclined cracks reduced the stiffness of the anchor block, which resulted in the lower slope of the load and displacement curve. The load, at which the inclined cracks occurred on the top surface of the anchor block, is defined as the cracking shear load of the anchor, $V_{cr}$.

While the inclined crack propagated to the edge of the top surface, another new vertical crack was generated from the top to the bottom on the front side of the anchor block. The vertical crack proceeded to a depth a little longer than the embedded depth of 240 mm. Then, the anchor reached its ultimate shear load, $V_u$. The inclined crack, which reached the edge of the block, was directed to the end tip of the vertical crack and thus formed into a semi-circular crack shape on the front surface of the block. Table 3 summarizes the cracking and ultimate shear loads, $V_{cr}$ and $V_u$, and the displacement at the maximum shear load, $d_{c,u}$ for the anchor specimens in the plain and SFRC.

**Table 3.** Summary of the cracking and ultimate shear loads and the displacement at the ultimate load obtained from the shear tests of anchors.

| Name of Specimen | $V_{cr}$ (kN) | | $V_u$ (kN) | $d_{c,u}$ (mm) |
|---|---|---|---|---|
| V000 | 29.6 | | 38.0 | 2.55 |
| | 30.3 | | 35.2 | 2.17 |
| | 40.5 | | 45.8 | 1.52 |
| | 23.6 | | 26.4 | 1.66 |
| | Mean | 31.0 | 36.3 | 1.98 |
| | (Std.) | (7.01) | (8.01) | (0.48) |
| V033 | 35.2 | | 42.9 | 3.78 |
| | 29.5 | | 38.2 | 3.81 |
| | 35.3 | | 41.8 | 4.06 |
| | 28.1 | | 30.5 | 2.76 |
| | Mean | 32.0 | 38.3 | 3.60 |
| | (Std.) | (3.78) | (5.63) | (0.58) |
| V067 | 34.8 | | 48.7 | 8.58 |
| | 34.6 | | 55.9 | 9.56 |
| | 33.4 | | 48.7 | 8.64 |
| | 29.8 | | 43.6 | 7.59 |
| | Mean | 33.1 | 49.2 | 8.59 |
| | (Std.) | (2.31) | (5.05) | (0.80) |
| V100 | 37.4 | | 52.1 | 12.27 |
| | 30.7 | | 59.4 | 9.72 |
| | 28.8 | | 53.8 | 13.76 |
| | 30.7 | | 43.7 | 9.56 |
| | Mean | 31.9 | 52.2 | 11.33 |
| | (Std.) | (3.77) | (6.48) | (2.04) |

Figure 8a shows a typical example of crack distributions of anchors in the plain concrete upon concrete breakout failure. The steel fibers mixed in concrete changed the crack distribution and shape of the concrete anchor under shear load. The anchors in SFRC showed a great improvement of the shear load and displacement resistance after cracking, while the strength of the anchor in the plain concrete quickly decreased and reached the final brittle fracture. This is because the steel fibers distributed in the concrete control the growth of the cracks and greatly improve the tensile capacity of the concrete block.

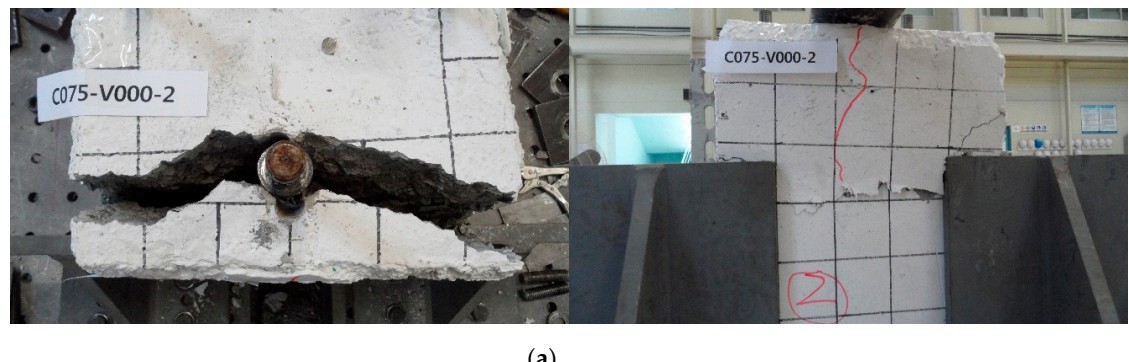

(**a**)

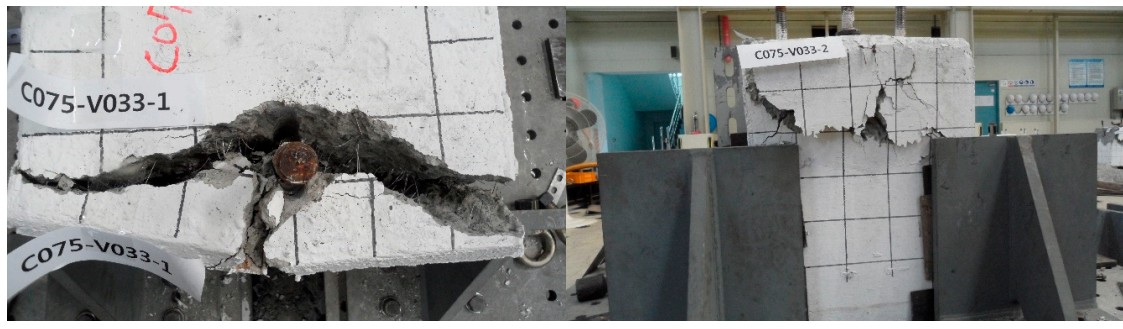

(**b**)

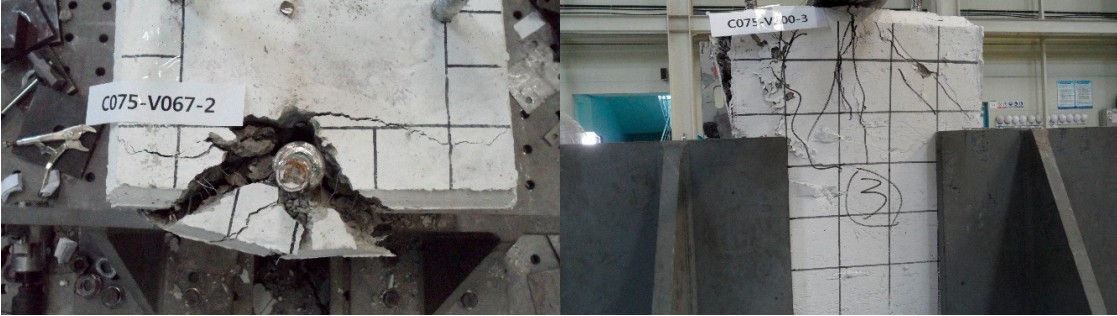

(**c**)

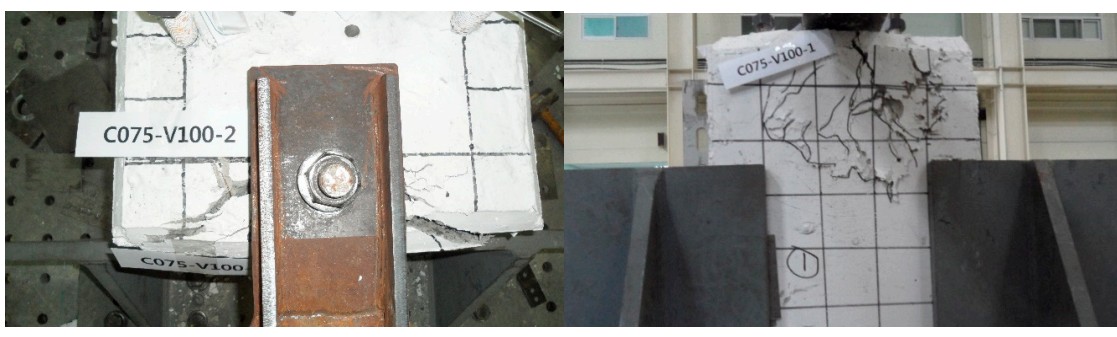

(**d**)

**Figure 8.** Crack distributions of the concrete anchor block upon concrete breakout failure: (**a**) plain concrete; (**b**) SFRC with $v_f$ = 0.33%; (**c**) SFRC with $v_f$ = 0.67%; (**d**) SFRC with $v_f$ = 1.00%.

Figure 8b–d show the crack initiation and propagation of the anchor installed in concrete reinforced with steel fibers. On the top surface of the block, additional inclined cracks occurred near the original inclined crack. The final fracture angle on the top surface tended to be lower than the 35° defined in the CCD method. On the front of the concrete block, another vertical crack was also generated along

the original vertical crack. The most distinctive feature in the anchors installed in the SFRC was the generation of a new type of radially straight crack originating from the center of the edge on the front side of the concrete block, which was independent of the previous inclined, vertical, and curved cracks. In particular, the anchor in concrete with higher steel fiber volume fractions showed more cracks and more complicated crack shapes.

## 4.2. Load and Displacement Resistance

The effect of steel fibers on the cracking and ultimate shear loads of the anchor was investigated ($V_{cr}$ and $V_u$, respectively). The linear regression analysis was performed with respect to the steel fiber volume percentages from zero to 1.00%. As shown in Figure 9, $V_{cr}$ is almost constant in the range of approximately 31.0 to 33.1 MPa, regardless of the increasing fiber volume fractions. However, $V_u$ increases almost linearly from approximately 36.3 kN in the non-reinforced concrete to 52.2 kN in the SFRC with a fiber volume percentage of 1.00%, which is approximately 1.44 times that of the non-reinforced concrete anchor. The slope of the line obtained from the linear regression analysis with fiber volume percentages of zero to 1.00% indicates an increase of approximately 17.6 kN in $V_u$ per 1% increase in the fiber volume percentage.

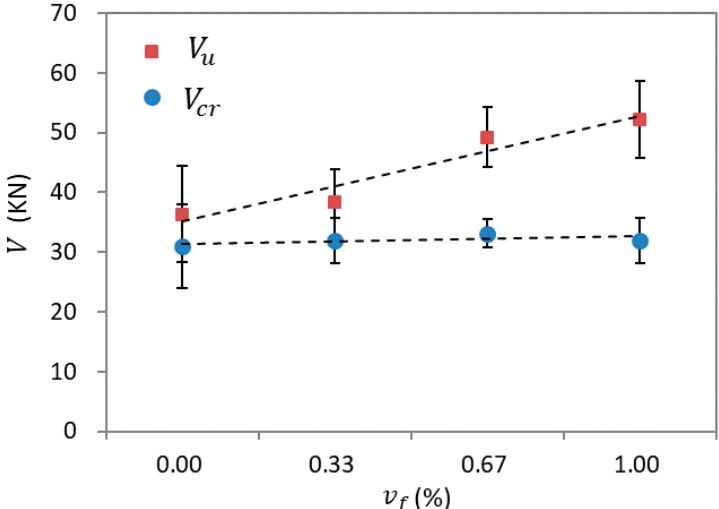

**Figure 9.** Cracking and ultimate loads measured from the shear tests of anchors with increasing fiber volume percentages from zero to 1.00%.

Figure 10 shows the displacement at the maximum shear load of the anchor, $\delta_{c,u}$, with respect to the steel fiber volume fractions. The linear regression line shows that the displacement of the anchor at the maximum load increases in proportion to the steel fiber content. The displacement of the anchor in the SFRC with a fiber volume percentage of 1.00% was approximately 11.4 mm at the maximum load, which is 5.74 times that of the non-reinforced concrete anchor. The slope of the linear line in the range of the fiber volume percentages from zero to 1.00% is approximately 9.92.

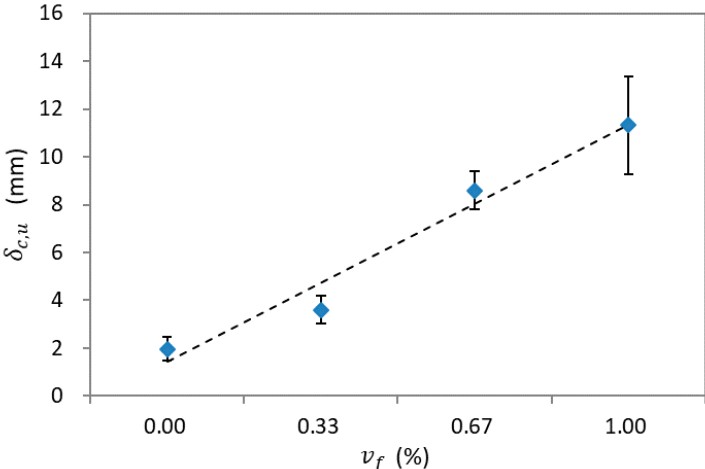

**Figure 10.** Displacement at the maximum shear load with increasing fiber volume percentages from zero to 1.00%**.**

### 4.3. Energy Absorption Capacity

This study also evaluated the energy absorption capacity of the plain and SFRC anchors, which can be used as an index for evaluating the fracture resistance of a material or structural member. Figure 11 shows the energy absorption and displacement curves of the plain concrete and SFRC anchors, which were calculated using the area of the load and displacement curve. The energy absorption increases almost linearly with increasing displacement, and the steel fibers significantly improve the energy absorption capacity. Since no criteria have been established for the energy absorption capacity of the concrete anchors, the evaluation methods of the energy absorption capacity in fiber-reinforced concrete beams have been reviewed.

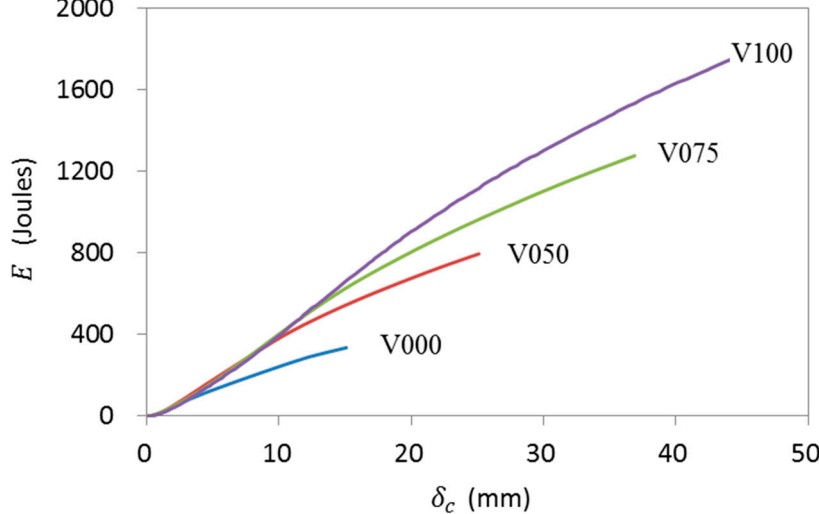

**Figure 11.** Typical energy absorption and displacement curves of the anchors in the plain concrete and SFRC.

Gopalaratman and Gettu [18] introduced the following three methods to evaluate the energy absorption capacity of SFRC beams: (1) the absolute energy absorption capacity until a specific deflection, (2) dimensionless indices related to energy absorption capacity, and (3) the energy absorption capacity by an equivalent flexural strength at a specified deflection in the post-cracking region. Among the three methods, Gopalaratman and Gettu [18] recommended the absolute energy absorption capacity at a certain deflection and the energy absorption capacity using the equivalent flexural strength. JSCE

SF-4 [38] adopted the absolute energy absorption method calculated at a deflection, and ASTM C 1550 [39], ASTM C 1609 [35], and BS EN 14,651 [37] adopted the evaluation method using the equivalent flexural strength at a specific deflection in the post-cracking region. The energy absorption evaluation method using the non-dimensional index was adopted in the ASTM C 1018 [40], but it is currently excluded from the ASTM recommendations.

Therefore, based on the absolute energy absorption method at a specific deflection, this study assessed the energy absorption capacity of anchors in plain concrete and SFRC. Two energy absorption values are defined: (1) the energy absorption until a displacement corresponding to the maximum shear load, $E_u$, and (2) the energy absorption from the displacement at the maximum shear load to a displacement when the post-cracking residual shear load reaches the level of the cracking shear load, $E_r$. Figure 12 shows $E_u$ and $E_r$ with increasing fiber volume percentages from zero to 1.00%. The linear regression lines show a linear increase of $E_u$ and $E_r$ with increasing fiber volume fractions. The $E_u$ of the anchors in the SFRC with steel fiber volume percentages of 0.33 and 1.00% are 120.3 and 518.5 J, which are approximately 2.64 and 11.4 times that of the anchor in the non-reinforced plain concrete, respectively. $E_r$ also increased from approximately 62.5 J in the plain concrete to 1387.2 J in the SFRC with a steel fiber volume percentage of 1.00%, which is 22.2 times that of the anchor in the non-reinforced concrete. The slope of the linear trend line obtained in the range of fiber volume percentages from zero to 1.00% is approximately 501.7 J per percent for $E_u$ and 1324.9 J per percent for $E_r$.

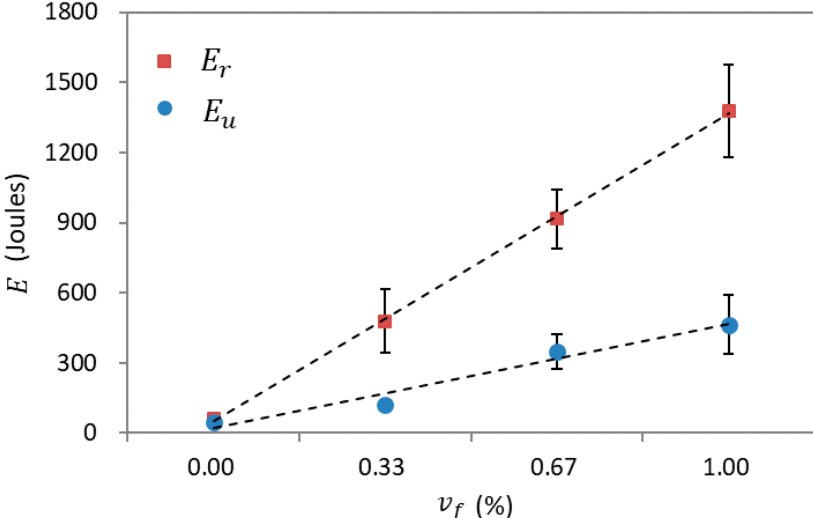

**Figure 12.** Energy absorption capacities $E_u$ and $E_r$ of the anchors in the plain concrete and SFRC with increasing fiber volume percentages from zero to 1.00%.

## 5. Relationship between Shear Behavior of Anchors and Tensile Performance of SFRC

The load and deflection curves of both the SFRC beams and anchors exhibited linear and elastic behavior before concrete cracking. This means that the tensile stress generated from the external loads is smaller than the cracking strength of the concrete and, thus, all of the materials including the concrete effectively endure the external loads. The inclusion of steel fibers has little effect on the cracking strength of the SFRC beams and anchors. As shown in Figures 3 and 9, the cracking strength of the beams, $f_{cr}$, and the cracking shear load of the anchors, $V_{cr}$, are almost constant regardless of the fiber volume fractions. On the other hand, the ultimate strength of the beams, $f_u$, and the maximum concrete breakout shear load of the anchors, $V_u$, increased almost linearly with increasing fiber volume fractions. According to the linear trend analysis, $f_u$ and $V_u$ increase by approximately 4.04 MPa and 19.90 kN per percent of fiber volume fraction, respectively.

The design shear resistance of anchors embedded in the non-reinforced plain concrete is dependent on the edge distance, embedded depth, and diameter of the anchor, as well as the compressive strength

of the concrete [8,9]. As mentioned, the steel fibers had little effect on the compressive and cracking strengths but greatly improved the ultimate flexural strength of the concrete. Thus, the concrete breakout shear resistance of the anchors needs to be determined using the ultimate flexural strength of the concrete, $f_u$, which can account for the improvement of the tensile capacity of concrete by the steel fibers rather than the compressive strength of the concrete specified in the ACI 318 design standards [7,8].

Figure 13 shows the relationships between the ratios of $f_u$ and $\sqrt{f_u}$ of the SFRC beams and $V_u$ of the SFRC anchors with the increasing fiber volume fractions to those of the non-reinforced plain concrete. $f_{u,o}$ is the ultimate flexural strength of the plain concrete beam, which corresponds to $f_{cr}$. $V_{u,o}$ is the concrete breakout shear resistance of the plain concrete anchor. $f_u$ tends to increase exponentially, while $\sqrt{f_u}$ and $V_u$ show linear increases as the fiber volume fraction increases. Furthermore, the rate of increase of $\sqrt{f_u}$ is very similar to that of $V_u$, which corresponds to the design equation expressed as a function of $\sqrt{f_c'}$, in which $f_c'$ is the compressive strength of concrete. Thus, based on the design equation specified in the ACI 318 [7,8], the concrete breakout shear resistance of an anchor in the SFRC can be expressed as

$$V_u = k\left(\frac{h_0}{d_0}\right)^2 \sqrt{d_0}\,\sqrt{f_u}\,(c_0)^{1.5} \tag{2}$$

where the influence factors of the anchor diameter, embedded depth, and edge distance are taken as those specified in the ACI 318 [7,8], and the term for the compressive strength of concrete, $\sqrt{f_c'}$, is replaced with a term for the ultimate flexural strength of concrete, $\sqrt{f_u}$, which includes the effect of the steel fibers on the tensile capacity of concrete. For the plain concrete anchor, $\sqrt{f_u}$ corresponds to the flexural tensile strength of concrete. The factor $k$ in Equation (2) can be determined by comparing the measured maximum shear load of anchors with the shear capacity calculated by Equation (2) without the $k$ term.

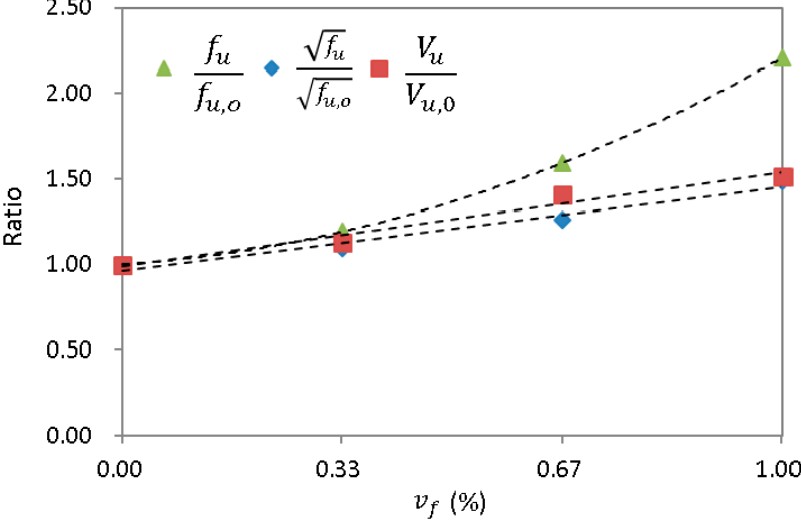

**Figure 13.** Ratios of the $f_u$ and $\sqrt{f_u}$ of the SFRC beams and $V_u$ of the SFRC anchors to those of the non-reinforced concrete with increasing fiber volume percentages from zero to 1.00%.

Figure 14 shows the variation in the factor $k$ with the change in the fiber volume percentages from zero to 1.00%. The average and standard deviation of $k$ are 3.83 and 0.19, respectively. Thus, $k$ is simplified as a constant equal to 4.0. Figure 15 compares the concrete breakout shear resistance of anchors calculated using the proposed Equation (2) with the measured average maximum shear loads. The calculated shear resistance shows very good agreement with the measurements. The differences between the calculated and measured shear loads are in the range of approximately 5–6%.

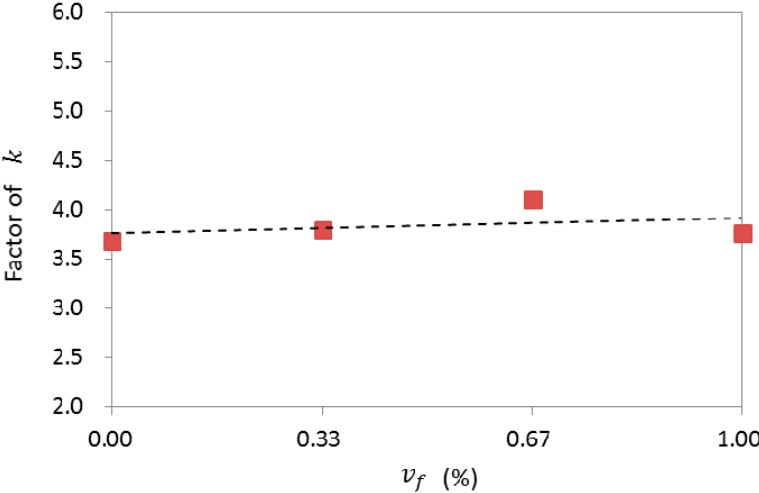

**Figure 14.** Variation in the factor *k* with the change in the fiber volume percentages from zero to 1.00%.

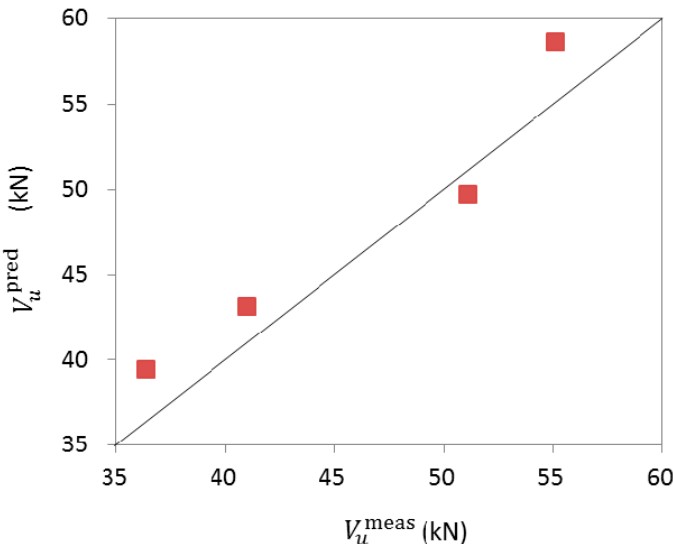

**Figure 15.** Comparison of the predicted and measured concrete breakout shear resistance.

In addition to the load capacity of an anchor, the energy absorption capacity can also be used to assess the fracture resistance of the anchor system. Therefore, this study evaluated the relationship between the energy absorption capacities of the anchor, $E_u$ and $E_r$, with the energy absorption capacity of the beam, $T_{150}$, which increases proportionally to the increase in fiber volume percentages from zero to 1.00%. Figures 16 and 17 show that the $E_u$ and $E_r$ of the SFRC anchors is proportional to the $T_{150}$ of the SFRC beams. Therefore, using the linear relationship with $T_{150}$, the energy absorption capacities $E_u$ and $E_r$ of an anchor can be determined as 3.63 and 9.97 times the value of $T_{150}$, respectively.

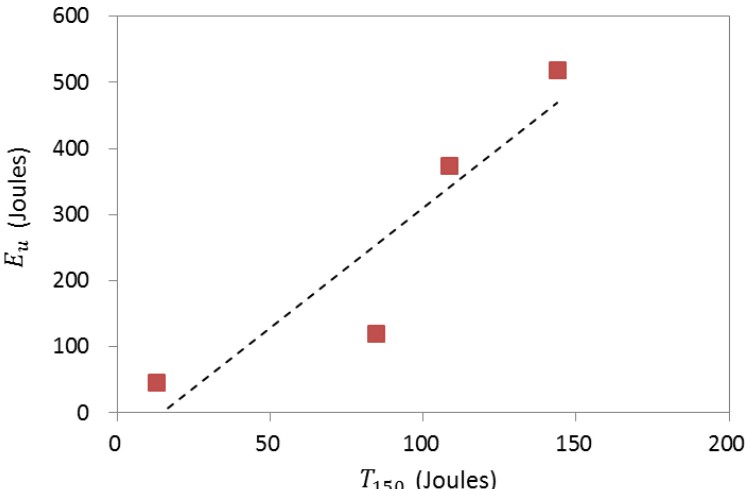

**Figure 16.** Relationship between the energy absorption capacity $E_u$ of the anchor and the energy absorption capacity $T_{150}$ of the beam.

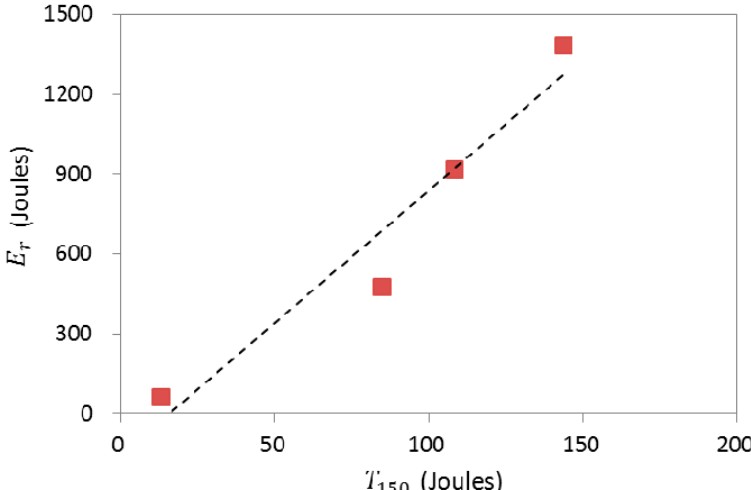

**Figure 17.** Relationship between the energy absorption capacity $E_r$ of the anchor and the energy absorption capacity $T_{150}$ of the beam.

## 6. Conclusions

The concrete shear resistance of an anchor is strongly dependent on the strength of concrete. The current design specifications are based on the compressive strength of concrete to determine the concrete breakout resistance of an anchor. The addition of steel fibers to concrete can greatly improve the tensile resistance of the concrete. Thus, this study investigated the effect of steel fibers on the shear failure mode and breakout resistance of anchors embedded in SFRC. Beam specimens were also prepared to assess the relationship between the tensile performance of SFRC beams and the shear resistance of SFRC anchors.

In the bending tests, the non-reinforced plain concrete beam showed a sudden decrease in the strength after cracking, which led to brittle failure. However, the beams reinforced with steel fibers showed deflection-softening or deflection-hardening behavior depending on the amount of steel fibers. The beams with fiber volume percentages of 0.67 and 1.00% continuously increased to reach the ultimate strength without decreasing the strength due to the cracking of concrete. The steel fibers had little effect on the cracking strength, but the ultimate flexural strength, post-cracking residual strength, and energy absorption capacity showed linear increases with increasing fiber volume fractions. The ultimate flexural strength $f_u$ and the residual strength $f_{150}^{eq}$ in the beam with a fiber volume percentage of

1.00% were approximately 2.21 and 1.70 times the cracking strength of the non-reinforced plain beam, respectively.

The shear tests of anchors also showed that the shear load and displacement capacities of the SFRC anchors increased almost linearly with the increase in the fiber volume fraction of steel fibers. The anchors in the plain concrete failed immediately after concrete cracking. The shear resistance $V_u$ in the SFRC with a fiber volume percentage of 1.00% was approximately 55.2 kN, which is 1.52 times that in the non-reinforced plain concrete. The displacement at the maximum shear load also increased by approximately 6.03 times in the SFRC with a fiber volume percentage of 1.00%. Thus, the energy absorption capacity at the maximum shear load for the SFRC anchors at fiber volume percentages of 0.33 and 1.00% was approximately 2.64 and 11.4 times those of the plain concrete anchor, respectively.

In the bending and shear tests, the ultimate strength of the beam and the maximum concrete breakout shear resistance of the anchor increased almost linearly with increasing fiber volume fractions. The design shear resistance of the concrete anchor is based on the compressive strength of concrete, which was rarely affected by the addition of steel fibers to the concrete with the fiber volume percentage less than 1.00%. Thus, based on the ACI 318 design equation, this study utilized a term for the ultimate flexural strength of concrete instead of the compressive strength to determine the concrete breakout shear resistance of an anchor in the SFRC. The calculated shear resistance of anchors in both the plain concrete and SFRC agreed well with the measured shear loads. In addition, the energy absorption capacity of the SFRC anchor with a fiber volume percentage of 1.00% showed a linear relationship with the energy absorption capacity of the SFRC beam, which increased proportionally to the increase in fiber volume percentages from zero to 1.00%.

**Author Contributions:** The authors J.-H.L., E.C. and B.-S.C. conceived and designed the study; In particular, methodology and formal analysis, J.-H.L. and B.-S.C.; experimental programming and investigation, J.-H.L. and B.-S.C.; writing—original draft preparation, J.-H.L.; writing—review and editing, E.C. and B.-S.C.; funding acquisition, E.C. All authors have read and agreed to the published version of the manuscript.

**Funding:** This research was funded by the National Research Foundation of Korea (NRF) grant funded by the Korea government (MSIT) (Project No. NRF 2020R1A4A1018826).

**Conflicts of Interest:** The authors declare no conflict of interest.

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
