# Peer review of "Shear Failure Mode and Concrete Edge Breakout Resistance of Cast-In-Place Anchors in Steel Fiber-Reinforced Normal Strength Concrete"

_applsci, doi:10.3390/app10196883_

Round 1

Reviewer 1 Report

The manuscript presents an important scientific issue related to the strength of anchors in concrete elements.
The paper presents the results of empirical tests of increasing the strength of concrete by adding steel fibers to it.
I believe that the article may be allowed for further processing after taking into account the amendments:

1. The test equipment used, types and parameters of sensors and aparatures, and their accuracy should be described in detail.

2. It is worth describing the choice of the quantity of steel fibers and their length in more detail. Was the selection made on the basis of the experimental design?

3. A greater number of articles from prominent scientific journals should be included in the references, which would increase the legitimacy of the selected amount of dispersed reinforcement used

4. It would be worth making a numerical model describing the functioning of anchors in concrete with different contents of steel fibers, which can be verified on the basis of empirical results.

Author Response

The manuscript presents an important scientific issue related to the strength of anchors in concrete elements. The paper presents the results of empirical tests of increasing the strength of concrete by adding steel fibers to it. I believe that the article may be allowed for further processing after taking into account the amendments:

1.   The test equipment used, types and parameters of sensors and apparatuses, and their accuracy should be described in detail.

(Response) The test measured the magnitude of the shear load using a load cell installed at the actuator. In addition, we have measured the displacement of the anchor using a LVDT installed on the back of the anchor. To ensure the accuracy of the sensors and the test procedure, the test was performed at one institute of KOLAS (Korea Laboratory Accreditation Scheme) that evaluates inspection, calibration, and tests in accordance with ISO.SCE Guide. We have described in more detail the setup of the specimen in the first paragraph of Section 2.3, the installation of the actuator with a load cell and LVDT in the second paragraph of Section 2.3. Please check the descriptions, marked in blue, in the Section 2.3 of the revised manuscript.   

 2.   It is worth describing the choice of the quantity of steel fibers and their length in more detail. Was the selection made on the basis of the experimental design?

(Response) As suggested by the reviewer, we have described the shape, length, and diameter of the steel fiber in the first paragraph in Section 2.1. The quantity of the steel fiber, corresponding to the fiber volume fractions of 0.33, 0.67, and 1.00%, was also described in Section 2.2. Please check the sentences, marked in blue, in the Sections 2.1 and 2.2 of the revised manuscript.  

 3.   A greater number of articles from prominent scientific journals should be included in the references, which would increase the legitimacy of the selected amount of dispersed reinforcement used

(Response) As recommended by the reviewer, we have included several articles from prominent scientific journals in the introduction section and references. Please check the introduction and references, marked in blue, in the revised manuscript.

4. It would be worth making a numerical model describing the functioning of anchors in concrete with different contents of steel fibers, which can be verified on the basis of empirical results

(Response) We have agreed with the reviewer’s comment. The results of this study showed that steel fibers had little effect on the compressive strength but greatly improved the ultimate flexural strength of concrete. Thus, we have replaced the compressive strength with the ultimate flexural strength of concrete to modify the current design equation. That is, this study proposed the modified equation as a function of edge distance, embedded depth, anchor diameter, and flexural tensile strength. The calculated shear resistance using the proposed Equation showed a very agreement with the measurements. The difference between the calculated and measured shear loads are in the range of approximately -5 to 6%. Please check the second, third, and fourth paragraphs of Section 5 in the revised manuscript, marked in blue.

Reviewer 2 Report

  1. In international literature and some codes "concrete breakout" often refers to axial loading. The authors may wish to use the term "concrete edge failure". 
  2. It is recommended to address the breakout angle in the tests, since failures in SFRC might deviate from the 35 degree, which is established for unreinforced concrete. 
  3. The authors may wish to also express the fibre content in kg/m3 which is widely used in fibre concrete specification worldwide.
  4. Some typos need to be corrected throughout the paper, e.g. : MPa (instead of MP) in ln 89 and elsewhere, kN in ln. 254, anchors in ln 422, macro-type in ln. 441.
  5. The tested anchors are of quite large size. Since the paper intends to generalise the findings for design purposes, the size effect of the failure needs to be discussed. Do the authors expect a proportional reduction of the shear resistance for e.g. shorter, 12 mm anchors? Or for anchors at a smaller edge distance?
  6. Did all failures occur in the concrete? Did the authors observe yielding or rupture of the steel bolts?
  7. The standard deviations of the test results in terms of maximum load and deformation at max. load must be noted. The results would ideally be summarised in a table together with the mean values to allow a better overview for the reader.  
  8. It would be of interest to elaborate on whether the square root of the tensile strength can generally replace the one of the compressive strength in the codes, also for plain concrete. Is this justified by the V00 test series?
  9. Previous research on the topic of anchors in fibre concrete is published but not referenced as background and basis to show this paper's novelty. It is recommended to add references on anchors loaded also in tension and failing under concrete breakout, which present many similarities to the shear investigations. Such axial and shear testing investigations have been carried out by P. Grosser, A Sharma, and B. Bokor in Stuttgart, Holschemacher in Leipzig, Schnell et al. in Kaiserslautern, Ayoubi in Frankfurt, Bergmeister et al. in Vienna. 
  10. Reference [11] is irrelevant to the statement in the paper and it must be substituted or removed. 
  11. Reference [17] discusses fibres with shape memory effect, which provide a different flexural behaviour as compared to the fibres used in this paper. This, and reference [18], should be removed, as also the statement ... The addition of steel fibers enhances flexural capacity in concrete beams... fiber content... is supported by more basic research references. 
  12. The authors should clarify the difference/novelty of the present research as compared to ref [3]. 

Author Response

  1. In international literature and some codes "concrete breakout" often refers to axial loading. The authors may wish to use the term "concrete edge failure".

(Response) Failure modes of concrete anchor under tensile loading are generally defined as steel failure, steel pullout, concrete breakout, concrete splitting, and concrete side-face blowout, while those under shear loading are defined as steel failure, concrete pryout, and concrete breakout. As commented by the reviewer, the term of concrete breakout failure is used under both the tensile and shear loading conditions. We have accepted the reviewer’s comment to change “concrete breakout” to “concrete edge failure” and “concrete edge breakout” including the title of the paper. Please check the revised term, marked in blue throughout the paper.

  1. It is recommended to address the edge angle in the tests, since failures in SFRC might deviate from the 35 degree, which is established for unreinforced concrete.

(Response) We have also found that the failure angle on the top surface of the specimen was generally lower than the 35 degrees. Moreover, what I have experienced, concrete edge failure angle even in the plain concrete was lower than the 35 degrees. As recommended by the reviewer, the future study is required to evaluate the mechanism of the failure angle somewhat lower than the 35 degrees, which has been defined in the design standards. Unfortunately, we have not measured the failure angle, so we have just discussed the fracture angle, which was generally lower than the defined 35 degrees and would be required as a future study. Please check the last paragraph of Section 4.1, marked in blue, in the revised manuscript.

  1. The authors may wish to also express the fibre content in kg/m3 which is widely used in fibre concrete specification worldwide.

(Response) As suggested by the reviewer, we have described the quantity of steel fiber, corresponding to the fiber volume fractions of 0.33, 0.67, and 1.00%, in the first paragraph of Section 2.2. Please check the sentences, marked in blue, in the Section 2.2 of the revised manuscript.

  1. Some typos need to be corrected throughout the paper, e.g. : MPa (instead of MP) in ln 89 and elsewhere, kN in ln. 254, anchors in ln 422, macro-type in ln. 441.

(Response) We have appreciated the reviewer’s comments. We have revised the typos and also finally checked the sentences and typos throughout the manuscript.

  1. The tested anchors are of quite large size. Since the paper intends to generalise the findings for design purposes, the size effect of the failure needs to be discussed. Do the authors expect a proportional reduction of the shear resistance for e.g. shorter, 12 mm anchors? Or for anchors at a smaller edge distance?

(Response) As commented by the reviewer, the anchor test generally requires a large size of specimen. This study defined the edge distance of the anchor as 75 mm (= 2.5 times the diameter of the anchor) to induce the concrete edge breakout failure. The inclined cracks generated on the surface of the block due to the shear load should not affect the other edge faces. Thus, the concrete block was designed to have a square top surface with a length of 500 mm. The embedded depth of the anchor in concrete is 240 mm, which is 8 times the diameter of the anchor. In addition, when the concrete anchor block was installed using steel angles at the four corners of the front and black sides at the bottom of the block. Thus, to prevent the reaction force of the steel angles from affecting the shear behavior of the anchor, the height of the block was defined as 680 mm in consideration of the embedded length of the block. The effects of the edge distance and embedded length of anchor on the concrete edge breakout failure are included in the original and proposed design equations.

  1. Did all failures occur in the concrete? Did the authors observe yielding or rupture of the steel bolts?

(Response) Yes, all the anchors showed the concrete edge failure mode. We did not observe any bending of the steel anchor during the tests. is because we have designed the edge distance and embedded length of the anchor in the specimens. In addition, when designing the specimens, we ensured the concrete edge failure as compared with the shear load, which could induce steel failure. 

  1. The standard deviations of the test results in terms of maximum load and deformation at max. load must be noted. The results would ideally be summarised in a table together with the mean values to allow a better overview for the reader.

(Response) As commented by the reviewer, we have included a table with the mean values and standard deviations for the cracking and ultimate load and the displacement at the ultimate load. Please check a new table, Table 3, in Section 4.1 in the revised manuscript, marked in blue. In addition, we have revised Figures 9, 10, and 12 to present the mean value and standard deviations in the figures.

  1. It would be of interest to elaborate on whether the square root of the tensile strength can generally replace the one of the compressive strength in the codes, also for plain concrete. Is this justified by the V00 test series?

(Response) The results of this study showed that steel fibers had little effect on the compressive strength but greatly improved the ultimate flexural strength of concrete. Thus, we have replaced the compressive strength with the ultimate flexural strength of concrete to modify the current design equation, which can be used in both the plain and SFRC concrete. The calculated shear resistance using the proposed Equation showed a very agreement with the measurements including the plain concrete anchor specimen. The difference between the calculated and measured shear loads are in the range of approximately -5 to 6%. Please check the second, third, and fourth paragraphs of Section 5 in the revised manuscript, marked in blue.

  1. Previous research on the topic of anchors in fibre concrete is published but not referenced as background and basis to show this paper's novelty. It is recommended to add references on anchors loaded also in tension and failing under concrete edge, which present many similarities to the shear investigations. Such axial and shear testing investigations have been carried out by P. Grosser, A Sharma, and B. Bokor in Stuttgart, Holschemacher in Leipzig, Schnell et al. in Kaiserslautern, Ayoubi in Frankfurt, Bergmeister et al. in Vienna.

(Response) As recommended by the reviewer, we have included several articles from prominent scientific journals in the introduction section and references. Please check the introduction and references, marked in blue, in the revised manuscript.

  1. Reference [11] is irrelevant to the statement in the paper and it must be substituted or removed.

(Response) Reference [11] ([14] in the revised manuscript) was referred to the Introduction section when explaining the concern over using the conventional steel bars, and also Section 3.1 when explaining the method of calculating the residual strength. Thus, we ask the reviewer to reconsider retaining the reference [11] ([14] in the revised manuscript). Instead, we have included several articles from prominent scientific journals in the introduction section and references. Please check the introduction, the first paragraph of Section 3.1, and references, marked in blue, in the revised manuscript.

  1. Reference [17] discusses fibres with shape memory effect, which provide a different flexural behaviour as compared to the fibres used in this paper. This, and reference [18], should be removed, as also the statement ... The addition of steel fibers enhances flexural capacity in concrete beams... fiber content... is supported by more basic research references.

(Response) We have deleted references [17] and [18] and their statements in the manuscript. Instead, we have included several articles from prominent scientific journals in the introduction section and references. Please check the introduction and references, marked in blue, in the revised manuscript.

  1. The authors should clarify the difference/novelty of the present research as compared to ref [3].

(Response) A previous study referred to [3] ] ([34] in the revised manuscript) assessed the relationship between the mechanical properties of SFRC and the shear resistance of anchors embedded in SFRC. Then it proposed a modified design equation that can calculate the concrete breakout strength of anchors in plain concrete and SFRC using the equivalent flexural strength ratio. SFRC exhibits similar compressive strength to plain concrete but a significant difference in tensile capacity. Thus, this study aims to expand knowledge on anchors in SFRC and replace the compressive strength of concrete with the tensile capacity, which can be more simply and generally employed for concrete anchors than the equivalent flexural strength ratio. We have clarified the difference and novelty of the present research in the introduction section. Please check the introduction section in the revised manuscript, marked in blue.

Round 2

Reviewer 1 Report

I accept the changes made by the authors to the manuscript.